## COMMENT

# Let's talk about race: changing the conversations around race in academia

Jasmine M. Miller-Kleinhenz [1✉], Alexandra B. Kuzmishin Nagy [2],
Ania A. Majewska [3], Adeola O. Adebayo Michael [4], Saman M. Najmi [2],
Karena H. Nguyen [3], Robert E. Van Sciver [5] & Ida T. Fonkoue[6,7]

Jasmine Miller-Kleinhenz et al. highlight the risk of science and academia's general neutrality to discussions around race and social justice. Their collectively-developed course represents a framework to begin these important discussions and improve conversations on race in academia.

The racial injustices experienced by Black Americans have existed for hundreds of years, yet the egregious murders of Black men and women in the spring and summer of 2020 ignited an awakening that shook our nation to its core, creating aftershocks that spared no sector of society, least of all our academic institutions. Many in the fields of science, technology, engineering, and math (STEM) have attempted to remain neutral to the issues of race in society, claiming that the scientific endeavor transcends race. However, there is well-established evidence for racial bias and name-based discrimination in recruitment[1–3], retention[1,2,4,5], hiring[1–3,5], and funding[6,7] in STEM. Therefore, it stands to reason that even if science *aims* to be objective, scientists that conduct scientific research are not always objective in matters of race.

The outcry from many Black scientists and their allies highlighted issues of racism in academia and resulted in the proliferation of movements such as #BlackInTheIvory[8], #BlackinX (e.g., BlackinChem, BlackinNeuro, BlackinPhysics, etc.)[9], and #ShutDownSTEM[10], among others. A growing number of academic departments and institutions issued statements expressing their desire to change the academic culture to create environments that are diverse, equitable, and inclusive (DEI)[11,12]. Many followed these statements with action by offering implicit bias training, forums and conversations focused on anti-racism, searches to increase diversity in hiring, and committees to assess departmental recruitment and retention practices[11]. While these activities are commendable and should be continued, these approaches are not particularly novel and are often implemented in ways that are unlikely to result in *lasting changes* in academic culture[13].

As postdoctoral researchers enrolled in an Institutional Research and Academic Career Development Awards (IRACDA) program supported by the National Institutes of Health (NIH) at Emory University, our training also includes teaching at historically Black colleges and universities (HBCUs). Because of our unique training environment, we experienced first-hand the need for long-lasting, cultural change on race issues in academia. After the traumatic murder of George Floyd and the social unrest that followed, we realized that we were all ill-equipped to discuss how racial inequalities impact and intersect with science in our conversations and

[1]Department of Epidemiology, Rollins School of Public Health, Emory University, Atlanta, GA, USA. [2]Department of Biochemistry, School of Medicine, Emory University, Atlanta, GA, USA. [3]Department of Biology, College of Arts and Sciences, Emory University, Atlanta, GA, USA. [4]Department of Pediatrics, School of Medicine, Emory University, Atlanta, GA, USA. [5]Department of Human Genetics, School of Medicine, Emory University, Atlanta, GA, USA. [6]Renal Division, Department of Medicine, School of Medicine, Emory University, Atlanta, GA, USA. [7]Research Service Line, Atlanta Veterans Affairs Health Care System (VAHCS), Decatur, GA, USA. ✉email: jmill37@emory.edu

interactions with our peers and students within our departments, labs, and classrooms. Additionally, while many of us were admitted into the program as underrepresented scientists in STEM, only a few of us identified as Black or people of color, highlighting that we all had different lived experiences and unique perspectives. Moreover, it became clear that our training thus far had overlooked the need to create a space for regular learning, discussions, and reflections on race inequalities in the US in general and particularly in academia.

Therefore, to address the gap in our training and to actively work towards lasting change for ourselves, our cohort designed a semester-long course on "Race and Social Justice in Academia." We wanted to create a course that, upon completion, would give us the tools to be effective educators at HBCUs in our immediate future, effective educators at any setting further along in our careers, and better citizens of society. Moreover, we hoped our course would provide other postdocs and faculty with a framework from which to discuss and implement actionable items that can impart meaningful changes in academia. Our goal with this commentary is to share the process of how we created and developed this course, in the hopes that it can serve as a framework for sustained efforts in other academic communities.

## The planning

**Community is key**. We reach out to collaborators, obtain feedback from colleagues, and network at scientific meetings when we hope to increase our research output; similarly, we must interact with others if we hope to improve our cultural competence and educate ourselves to become anti-racist[14,15]. In our case, we were fortunate to have an existing community that was formed from our common postdoctoral funding, awarded from the same institution in the same year. As a diverse community of scientists, we recognized that, while we felt sensitive to racial injustices, we still had much to learn. Building on this cohort model, we took the initiative to become anti-racist and to become better people, and thus better scientists, educators, and colleagues.

The successful outcomes of our initiative can be directly attributed to the community we created in our cohort. While we were sorted into our cohort because of the NIH IRACDA program, we are surrounded by various communities. A community can be formed within a department, a lab, or a professional organization. When diversity exists within these communities, it is a strength. Diversity can come from one's race, ethnicity, gender, disability, national origin, age, health status, religion, sexual orientation, gender identity, gender expression, socio-economic standing, immigration status, or family background. Depending on your situation, your community may not be diverse in some of these areas, but how these categorizations intersect for the members of your community will inform diverse perspectives and lead to better outcomes. Moreover, it is not necessary to have all aspects of identity to build empathy towards people underrepresented in academia. It is important to consider discussing what identities may be missing from your community to generate further discussion.

**Have a collective conversation**. Regardless of our community composition, we found it essential to have a collective conversation to discuss how we would educate ourselves about the topics of race, racial inequality, and social justice (Fig. 1A). In our initial meetings, we shared our own personal objectives, motivations, and goals for engaging in such a course. This process highlighted common interests and led to fruitful discussions about key aspects that were missing from the original course design. Most importantly, this process ensured that underrepresented members of the group were not shouldering the burden of finding,

suggesting, and outlining the scope of discussion. Simply put, the goal of these discussions was to avoid the burden that often falls on Black, Indigenous, and people of color (BIPOC) to initiate and lead similar conversations in academia at the undergraduate, graduate student, postdoctoral, and faculty levels[12,16,17]. At the end of this conversation, everyone agreed on the major topics and learning objectives (Table 1), and equitably distributed the work of leading future sessions.

**Meet regularly**. Similar to regular lab meetings that improve our research, becoming anti-racist is a lifelong commitment. Following the collective conversation mentioned in the previous section, we felt that our cohort needed an ongoing, multi-week discussion to examine the root of the racial injustices that we had recognized as academics and to discuss possible strategies to mitigate these injustices. For our cohort, we chose to meet over the course of 16 weeks to give each other the space to share our experiences and ideas and to ensure accountability in participation and facilitation. Specifically, we chose a weekly meeting time that aligned with everyone's schedule and assigned two different members of the cohort to lead sessions each week. This structure gave individuals the opportunity to choose a week where they could invest more time in preparing and facilitating a session while still keeping everyone involved. We recognize that this process will require extra effort and time, but the reality is that there is no quick fix to dismantling systems of oppression within our institutions and society at large. A commitment to anti-racism requires a conscious, intentional, and continuous effort to achieve lasting changes in our mindset. These changes may then lead to solutions that are realistic and implementable within our communities (Fig. 1B).

## The implementation

We used backward design principles of pedagogy[18] to determine course learning objectives as a group (Table 1). Each participant signed up to serve as a facilitator for two learning objectives, which resulted in two people co-leading each session. Pairs collaborated to outline learning outcomes (or sub-objectives), create activities, or implement existing pedagogical techniques and tools to complement those outcomes. In addition, speakers were invited, such as educators from HBCUs and experts of inclusive pedagogy, to contribute to the conversation by sharing their lived experiences.

We employed active, student-centric, inclusive teaching methods[19–22] to share content and engage all participants in the learning experience. For example, we often used think-pair-share activities[23], during which participants were offered time to reflect, then collaborated with a partner, before sharing their responses with the rest of the group. We also constructed activities modeled after the "jig-saw" method[24], where participants were arranged into different groups with distinct tasks. These groups were then disassembled and new groups, composed of participants with different original tasks, were formed. The members of these new groups then took turns sharing their tasks and responses with the other members until everyone was exposed to all the tasks and responses. All activities were inspired by primary literature[25], existing resources[12,26,27], and relevant news articles[28].

Our overall strategy was to have interactive sessions that encouraged an exchange of ideas. To promote such discussions, we often posed open-ended questions to invite *all* participants' voices to be heard[29]. In addition, trauma-informed pedagogical practices were applied to ensure that participants were not re-traumatized (e.g., provided trigger warnings for potentially graphic content[30]). Interactions as facilitators and participants allowed us to build trust and rapport as well as break through

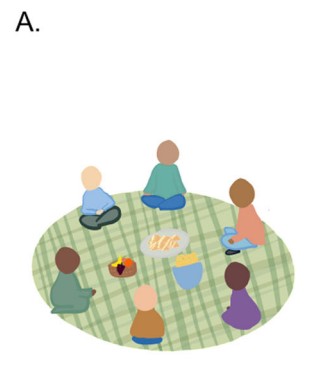

A.

Step One:
participate in ongoing conversations
with your peers, listen and learn

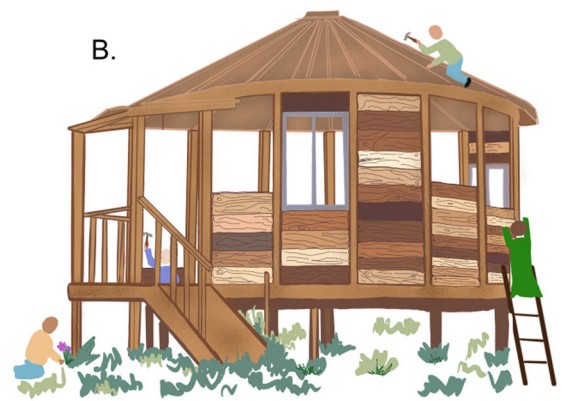

B.

Step Two:
work together to build an academic
environment that is welcoming

**Fig. 1 A model for building an equitable environment for academics through conversation.** Following our model, groups everywhere can use our course as a guide to (**A**) facilitate conversations around social justice and come to an understanding about the changes they can make to achieve racial justice and equity in academia, and (**B**) actually make those lasting changes. On the left-hand side, a diverse group of people engages in a conversation while having a picnic. On the right-hand side, a diverse group of people works together to build a home representing academia, with different colors of wooden planks, representing the many skin tones that are present in academia.

some of the discomfort that comes with these conversations. We also included debriefings and opportunities to reflect as a group on current events, recent activities, personal attitudes, and perceptions of societal issues.

## Our guiding principles

At the conclusion of our course, we reflected on what allowed us to have such productive, sustained conversations. We determined that during our conversations, there were key guiding principles that led us to have discussions that resulted in a change in our hearts, minds, and actions (Table 2).

## Reflections and broader impacts

**On ourselves**. This course had a direct impact on how we, as the students, viewed discussions about race. Prior to participating in this course, many fellows mentioned experiencing acute discomfort and hesitancy in engaging in conversations centered on race. One fellow reflected, "I recently realized how far I've come when I saw how uncomfortable my white colleagues were when discussing race. Similar to me before taking the course, they didn't want to say the wrong thing and feared offending someone or facing their white privilege." After this course, participants gained the confidence and courage to overcome this discomfort and engaged in these conversations with colleagues, friends, and family. The course also created a space for fellows to extend conversations about race with diverse groups of all backgrounds, whereas before, "I predominantly discussed matters of race and academia with other people of color. The academic environments I had been a part of had made me feel that discussions around race were either inappropriate, irrelevant, and/or unwelcome… Being able to bring these pertinent conversations to the fore and making plans for lasting change within the classroom, research group, department, and the university has shifted my perception of academia. I am starting to feel welcome and that I belong here too." The course has also allowed fellows to confront microaggressions and other injustices within academic spaces, such as having "the courage to call in a mentor/director on issues of scientific neutrality with regards to race and social justice. It took a lot of courage to overcome the fear that I might be penalized for it in some way, directly or indirectly. But I have come to learn that my active voice is needed and valued in the space that I

occupy. As a Black woman in academia, I found the courage to speak up and speak out." Lastly, this course allowed for fellows to see from diverse perspectives, with one fellow indicating the course "has significantly developed and improved my patience towards colleagues, students, and friends who have different views than mine on racial injustice and social inequities in the United States."

**On our students**. We have also taken these experiences back to our classrooms. Two fellows, with feedback from cohort members, included diversity statements in their syllabi and intentionally read them aloud on the first day of class to demonstrate their commitment to fostering inclusive and equitable learning environments. It also motivated other fellows to address barriers minority students face in STEM. One fellow "dedicated class time to talk about Black scientists for Black history month and to acknowledge the lack of representation in academia which stems from the challenges Black Americans face in the sciences" and another became "comfortable addressing the historic whitewashing of science in the chemistry and biochemistry classes I taught at an HBCU. Celebrating and highlighting the contributions of diverse peoples to science has become a key aspect of my inclusive teaching and mentorship."

**In our academic communities**. Lastly, the impact of participating in this course extended beyond its impact on us as individuals, to our departments and training programs. One fellow brought the cohort-based model of discussing race and social justice to enhance his department's ongoing DEI initiatives. Another fellow served as a trainee on a faculty search committee and worked with a graduate student to develop a rubric by which to assess candidates' DEI statements. There was initial resistance to having trainees serve on the search committee and a reluctance to using a rubric to weigh DEI statements as part of the faculty application package. However, following the search, the Chair of the Department stated:

All of us hold opinions that are shaped by our past experiences and, certainly, a trainee would never have been put onto a Search Committee for new faculty when I was either a graduate student or a postdoctoral fellow. For our recent faculty search, initially, my fear was that the inexperience of trainees would make their

**Table 1 Course learning objectives were agreed upon by the group.**

| Course learning objective | Lesson sub-objectives |
| --- | --- |
| Becoming comfortable having uncomfortable conversations surrounding race and social justice within our own group. (see Supplementary Note 1. Supplementary Table 1) | 1. Discuss how people may take comfort in their own privilege and how that influences their ability to discuss race, social justice, and action. 2. Define intersectionality, identify examples of intersectionality, and connect this concept to conversations of race. 3. Examine and describe how language and semantics are important in these conversations. 4. Practice engaging in conversations about race. |
| Deepening our understanding of race as a construct: perceptions on race throughout history and the nature of anti-Black racism in the U.S. (see Supplementary Note 2, Supplementary Table 2) | 1. Describe African American history, including milestones and lingering systems of anti-Black racism in the United States. 2. Understand the African Diaspora and how the trans-Atlantic slave trade cemented race as a construct in the United States. 3. Understand how BIPOC contributed to the labor movement and how this contribution disproportionately benefited white Americans. |
| Understanding our personal relationship to white supremacy and anti-Black racism. (see Supplementary Note 3, Supplementary Table 3) | 1. Define and recognize white supremacy. 2. Gain the understanding that we all have a relationship with white supremacy, and learn to identify the intersectionality of our experiences. 3. Acknowledge and accept past behavior and learn to move forward. |
| Evaluating evidence-based practices for mentoring and increasing the retention of BIPOC in STEM. (see Supplementary Note 4, Supplementary Tables 4-5) | 1. Identify the historical and social constructs that impede the scientific success of BIPOC at various levels of education. 2. Evaluate current programs that attempt to address these inequities at various levels of education. 3. Brainstorm ways in which to improve programs that increase retention and representation of BIPOC in STEM at the undergraduate level. 4. Design a framework from which to foster inclusivity in the classroom and pass on (academic) cultural capital to our undergraduate students. |
| Investigating the impacts of macro- and micro-aggressions in the classroom and beyond. (see Supplementary Note 5, Supplementary Table 6) | 1. Easily differentiate between microaggression and macroaggression and become familiar with their different forms or types. 2. Become familiar with the different categories of racial microaggressions. 3. Explain the roots of microaggression and macroaggression. 4. Articulate the impact of repeated (lifetime) macro- and microaggressions on behavior, growth, and achievement of students and colleagues. 5. Quickly recognize and address microaggressions when you witness or cause one. 6. Appropriately respond to a microaggression and make micro interventions—short-term or transient interventions without a counselor. |
| Implementing social justice practices in the higher education curriculum. (see Supplementary Note 6, Supplementary Table 7) | 1. Discuss how recent movements (e.g., #BlackInTheIvory, #BlackAFInSTEM, #PublishingPaidMe) have spurred more awareness about barriers BIPOC scientists and writers face. 2. Evaluate methods for implementing anti-racist practices in our labs, classrooms, departments, and colleges. 3. Brainstorm ways in which to increase transparency in negotiation proceedings, hiring practices, etc. in higher education. |
| Recognizing, alleviating, and dismantling local systems of oppression. (see Supplementary Note 7, Supplementary Table 8) | 1. Explain the concepts and theories of oppression. 2. Define social justice, economic security, and equality. 3. Identify the role(s) played by human behavior in the social environment. 4. Explore personal biases and stereotypes that can affect human behavior. 5. Identify the impact of privilege and oppression and the potential power dynamics of race in the context of how the dynamics of oppression impact the human developmental process. 6. Identify action strategies used to address and dismantle oppression. 7. Identify ways individuals, social movements, and institutions can promote justice and equality and alleviate oppression. |
| Fostering anti-racist behavior among our students, academic peers, and community organizations. (see Supplementary Note 8, Supplementary Table 9) | 1. Define anti-racism. 2. Recognize why we need to take personal responsibility for eliminating racism. 3. Construct your action plan as an anti-racist in your family, department, and classroom. 4. Make a plan on how to continue your journey of becoming anti-racist as an individual and part of an organization. |

These were the outcomes we hoped to gain by participating in the course. Each pair of facilitators created Sub-Objectives specific to each lesson/class meeting. Related to Supplementary Notes 1–8 and Supplementary Tables 1–9.

**Table 2 Guiding principles for having productive, long-term conversations around issues of race and social justice in academia.**

| Conscious effort leads to sustainable action and change. | Dismantling racism in academia is not a quick fix. Anti-racism requires a time commitment and willingness to work towards a lasting change. |
|---|---|
| Building trust allows for open and honest conversations. | Everyone in the group must feel psychologically safe in order to freely share their feelings without fear of retribution. Conversations around issues of race and social justice can start within small groups and cohorts, which helps provide the foundation and trust needed for the difficult and honest conversations throughout the course. In academia, there are already such "cohorts" or bonded groups in the form of departments, labs, committees, class groups, etc. |
| Being comfortable having uncomfortable conversations facilitates growth. | Conversations about race are not easy. Often, we dampen or avoid the ugly truth to maintain a semblance of harmony or to hide personal biases. However, tackling institutional racism is impossible without first recognizing and tackling our own biases, acknowledging that people are the source of widespread institutional racism, and all people have the capacity to change and do better. |
| Active listening allows others to feel that they are seen, and their experiences are heard. | Listening is not just waiting to speak; it is hearing what others have to say and giving them the space to share their experiences without being dismissive or inserting our personal ego. This includes framing questions and discussions in ways that respect others to prevent gaslighting[31]. |
| Empathy towards others' experiences validates their lived experience. | You may have never experienced racism, but you can empathize with those who have. Avoid dismissing the experiences of the oppressed or arguing on behalf of the oppressor. |
| Diversity is a strength. | Diversity within the group will lead to the diversity of thoughts. Given that the topics for discussion/reflection are decided by the group, the more a group can leverage its diversity, the more meaningful the conversations will be. |
| Engaging and becoming agents of change advances anti-racism everywhere. | While conversations around race may initially take place within a cohort, the ultimate goal is to extend them into our various communities. Conversations on race, equity, and social justice can be uncomfortable and the natural reaction is to withdraw and disengage. However, constant engagement, and encouraging the same engagement from peers, students, friends, and family will allow all of us to move past the discomfort and into growth. This domino effect will help spur others into action. |

opinions of limited utility. Furthermore, I thought that trainees would be put into a tough spot if they found themselves at odds with the prevailing opinion of faculty on the Search Committee, so part of my reluctance was driven by my desire to protect our trainees. Nonetheless, I agreed to give it a test, partly at the urging of my younger colleagues, and both of my reservations proved to be unfounded. I saw our trainees rise to the challenge to provide both valuable input and a useful perspective that would have been missing had they not participated. This included the DEI rubric they developed that provided a useful framework for evaluating candidate DEI activities in a reasonably objective fashion, focusing attention on actual accomplishments and not just words.

Our course also impacted the Fellowships in Research and Science Teaching (FIRST) IRACDA program at Emory, which has a primary mission of fostering relationships with partner minority-serving institutions. Despite this goal, the training did not include explicit preparation in cultural competence and race and social justice. Therefore, we pushed for this course to be formalized as a requirement for future FIRST fellows through numerous conversations with FIRST administration. Similar to the reluctance that faculty had about trainees serving on a search committee, there was hesitancy to formally adopt this cohort-based approach to addressing race and social justice in academia. One co-director of FIRST acknowledged:

While the leadership of FIRST was a bit skeptical about a fellow-designed and led course at the outset, the value to the individual participants and to the program is now clear. We have formally added this course as part of the permanent FIRST curriculum, however, the impact is broader than one program. As a leader of multiple programs, I plan to build on this course either by asking FIRST fellows to provide specific modules or by asking them to work with other trainees to implement their own personalized course. Such an approach allows trainees to take ownership and personalize their experiences. The process of

designing and teaching themselves also strengthens the cohort, which is key to building an inclusive environment.

Before this class, conversations on the topic of the race would make many of us uncomfortable, often afraid to speak up in fear of saying something wrong. This course helped instill confidence and courage in engaging in these conversations with colleagues and family. Participants in the course now have the competence to facilitate and partake in these discussions in the future.

## Conclusion

As a scientific community, we have reached a decisive moment to make a significant contribution towards achieving improved racial justice and equity in academia. Researchers and academics, who are interested in social equity, can no longer afford to just remain in our research silos or teach in our areas of specialization. Each of us have a responsibility to dismantle racism, in all forms, within our labs, classrooms, departments, institutions, and various communities. As a cohort, we recognized the need for a sustainable change in our training, agreed to take action, and committed to the process. As a result, members of our cohort reported that this class gave them the tools and also the courage to speak up against any form of witnessed injustice. Meanwhile, others reported including intentional discussions about racial injustices in their classes and becoming more in-tune with their students' experiences. We were fortunate to have members of the group who could speak on racial injustice from experience. However, we believe that conversations around race and social injustice can and should take place in every community, regardless of their racial make-up. Having ongoing conversations focused on race and social justice within academia is a vital step towards creating anti-racist environments that are conducive to diverse and talented scientific communities.

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

## Acknowledgements

We want to warmly thank Drs. Mark Lee, Lisa Hibbard, and DeShawn Preston for their insights during the conceptualization of the course and writing of this manuscript. This manuscript would not have been possible without our mentors and students at Spelman College and Morehouse College. We want to thank Drs. Steven L'Hernault and Anita Corbett for providing quotes for the reflection. We are also thankful to the Emory Institutional Research and Academic Career Development Award (IRACDA) leadership and alumni. JMMK, ABKN, AAM, AOAM, SMN, KHN, and REVS were funded by the National Institutes of Health (NIH) IRACDA NIH/NIGMS (2K12GM000680). REVS was funded by NIH F32 (DK127848). ITF was funded by NIH T32 (DK-00756).

## Author contributions

JMMK and ITF conceived of this anti-racism framework as a peer-taught class. JMMK and SMN conceived of the Figure and SMN created it. JMMK, ABKN, AAM, AOAM, SMN, KHN, REVS, and ITF contributed equally to planning and syllabi (supplemental information). JMMK, ABKN, AAM, AOAM, SMN, KHN, REVS, and ITF wrote the manuscript, which was reviewed, revised, and improved by all authors. JMMK, ABKN, AAM, AOAM, SMN, KHN, REVS, and ITF read and approved the final manuscript.

## Competing interests

The authors declare no competing interests.
