## [Transparent Peer Review File · Communications Biology]

Reviewers' comments:

Reviewer #1 (Remarks to the Author):

I appreciate the manuscript and the approach to collective and individual learning that it reports. After my first read, it stayed with me and I found myself thinking about it and you. Many readers will not have thought about a course design and implementation as a learning approach for themselves. In that regard, it is novel and yet, you present an approach to learning that faculty recognize. A course could be developed in any number of ways and your design has rigorous learning objectives and sub-objectives. Additionally, it enabled personal/individual reflections and perspectives as well as historic, cultural and disciplinary considerations.

The manuscript can stand as a report of a collectively developed course about race and racism or it can be expanded to provide an example of and, an exploration of, a type of learning that can enable lasting change in faculty and perhaps, lasting changes of academic culture. My comments about strengthening the paper invite reflections the learning that occurred and the potential of this framework to contribute to changes in academic culture.

Re implementation and guiding principles: What did you experience and learn as the course was implemented? What emerged that you didn't anticipate? Were the guiding principles developed prior to course implementation or where they a product of this project? In what ways are you changed by this experience?

The observation about creating lasting changes in academic culture (lines 38-40) leapt off the page. How is this learning project linked to creating lasting changes in academic culture?

Lines 50-59 are central. What more can you say about contributing to lasting changes in academic culture?

Consider including a "results/reflection" section ahead of the conclusion and return to these thoughts. What are you now prepared to bring to your students and to your faculty colleagues? What research questions have emerged that could contribute to creating departmental environments that welcome discussions of race, racism, and anti-racism?

Several ideas in the conclusion seem to be reflections on the learning project. Following that part of the discussion, you could then make concluding remarks.

Reviewer: Mary E. Boyce, PhD, Asst. Vice Provost
University of Michigan

Reviewer #2 (Remarks to the Author):

The manuscript by Miller-Kleinhenz et al. is timely, and is well-written. A strong case is made for having structured discussions about race in a group of trusted colleagues, to prepare for discussions about race in other, less controlled settings. This is logical and though not revolutionary, is a great idea, and seems to have worked for this group of postdoctoral fellows. The Emory IRACDA fellows put significant effort into designing a curriculum, which encompassed 16 weeks. This manuscript shows great enthusiasm and commitment on the part of the fellows.

There are two areas where the manuscript could be improved. Ideas for improving the manuscript are suggested in the next two paragraphs.

It would be interesting to know whether and how this endeavor affected others at Emory, perhaps those less committed to understanding race, or less interested in spending time in such discussions. Are there any anecdotal or measurable effects on others outside the participant group who met for the intensive discussion over 16 sessions? Similarly, are there any measurable ways or anecdotes concerning how the fellows used this preparation to engage in discussions in the classroom or with faculty, staff and students outside the class?

Finally, could more details of the curriculum be shared so others would not need to recreate it? Perhaps this could be done on a website or other venue. This could save others time. For example, it might be possible to have other IRACDA programs test the various segments of the curriculum, and for it to be evaluated. How does the curriculum intersect (or does it) with various units available through CIMER?

We thank our reviewers and editor for the helpful comments and suggestions. We have responded to the comments with additions to our manuscript as described below. We collectively feel that these suggestions have strengthened our manuscript, and hope you feel the same.

Reviewers' comments:

Reviewer #1 (Remarks to the Author):

I appreciate the manuscript and the approach to collective and individual learning that it reports. After my first read, it stayed with me and I found myself thinking about it and you. Many readers will not have thought about a course design and implementation as a learning approach for themselves. In that regard, it is novel and yet, you present an approach to learning that faculty recognize. A course could be developed in any number of ways and your design has rigorous learning objectives and sub-objectives. Additionally, it enabled personal/individual reflections and perspectives as well as historic, cultural and disciplinary considerations.

The manuscript can stand as a report of a collectively developed course about race and racism or it can be expanded to provide an example of and, an exploration of, a type of learning that can enable lasting change in faculty and perhaps, lasting changes of academic culture. My comments about strengthening the paper invite reflections the learning that occurred and the potential of this framework to contribute to changes in academic culture.

Response: We thank the reviewer for their interest and enthusiasm for our work.

Comment #1

Re implementation and guiding principles: What did you experience and learn as the course was implemented? What emerged that you didn't anticipate? Were the guiding principles developed prior to course implementation or where they a product of this project? In what ways are you changed by this experience?

Response: We thank the reviewer for their thoughtful questions around our implementation process as well as our guiding principles. We have included a new section with Reflections and Broader Impact which expands on our experiences with the implementation of the course, how it impacted us as individuals, and how we are changed through personal anecdotes. Our guiding principles were consolidated upon reflection at the end of the course which we have now noted in the "Guiding Principles" section of the manuscript.

Page 6: Guiding Principles "At the conclusion of our course, we reflected on what allowed us to have such productive, sustained conversations. We determined that during our conversations, there were key guiding principles that led us to have discussions that resulted in change in our hearts, minds, and actions (Table 2)."

Page 7: see new section "Reflections and Broader Impacts"

Comment #2

The observation about creating lasting changes in academic culture (lines 38-40) leapt off the

page. How is this learning project linked to creating lasting changes in academic culture?

Lines 50-59 are central. What more can you say about contributing to lasting changes in academic culture?

Response: We agree with the reviewer that the ultimate goal of this learning project would be create lasting change in academic culture. We recognize that seeing lasting change will require years of observation. However, we are seeing direct changes as a result of taking this course that are indicative of lasting change. One of our co-authors sat on a departmental faculty search committee and implemented a DEI rubric that as instrumental in guiding decisions made by the committee. Additionally, this course will now become a permanent part of the Emory FIRST-IRACDA curriculum for future incoming cohorts of postdoctoral fellows. In response to the reviewer's comments, we have added a section in the manuscript highlighting these observed changes.

Page 7: see new section "Reflections and Broader Impacts"

Comment #3

Consider including a "results/reflection" section ahead of the conclusion and return to these thoughts. What are you now prepared to bring to your students and to your faculty colleagues? What research questions have emerged that could contribute to creating departmental environments that welcome discussions of race, racism, and anti-racism?

Response: We thank the reviewer for their suggestion, and we agree that a "reflection" section would allow for us to share reflections on what we are prepared to bring to our students and our colleagues. We have included the recommended section.

Page 7: see new section "Reflections and Broader Impacts"

Comment #4

Several ideas in the conclusion seem to be reflections on the learning project. Following that part of the discussion, you could then make concluding remarks.

Response: We have concluding remarks within a "Conclusion" section following the "Reflections and Broader Impacts".

Reviewer: Mary E. Boyce, PhD, Asst. Vice Provost
University of Michigan

Reviewer #2 (Remarks to the Author):

The manuscript by Miller-Kleinhenz et al. is timely, and is well-written. A strong case is made for having structured discussions about race in a group of trusted colleagues, to prepare for discussions about race in other, less controlled settings. This is logical and though not revolutionary, is a great idea, and seems to have worked for this group of postdoctoral fellows. The Emory IRACDA fellows put significant effort into designing a curriculum, which

encompassed 16 weeks. This manuscript shows great enthusiasm and commitment on the part of the fellows.

Response: We thank the reviewer for their thoughtful review of our work.

There are two areas where the manuscript could be improved. Ideas for improving the manuscript are suggested in the next two paragraphs.

Comment #1

It would be interesting to know whether and how this endeavor affected others at Emory, perhaps those less committed to understanding race, or less interested in spending time in such discussions. Are there any anecdotal or measurable effects on others outside the participant group who met for the intensive discussion over 16 sessions? Similarly, are there any measurable ways or anecdotes concerning how the fellows used this preparation to engage in discussions in the classroom or with faculty, staff and students outside the class?

Response: We agree with the reviewer that it would be helpful to understand the broader impact of our course on others at Emory. We have included broader impacts in a new "Reflection and Broader Impact" section of the manuscript. Within the new section we include our (the authors) own anecdotes as well as anecdotes from those at the departmental and program leadership level that could speak to some measurable change.

Page 7: see new section "Reflections and Broader Impacts"

Comment #2

Finally, could more details of the curriculum be shared so others would not need to recreate it? Perhaps this could be done on a website or other venue. This could save others time. For example, it might be possible to have other IRACDA programs test the various segments of the curriculum, and for it to be evaluated. How does the curriculum intersect (or does it) with various units available through CIMER?

Response: We thank the reviewer for offering the suggestion that we share details of our curriculum so that others might be able to utilize our materials when developing their own course. We have created lesson plans, including materials such as presentations and activities, for each of our classes, and will add them as supplemental materials to this manuscript. We hope these materials will guide and inspire others to take part in productive and ongoing conversations in their communities. In addition, as the reviewer suggested, we will be sharing our insights with other IRACDA programs during the annual IRACDA meeting this summer through an interactive workshop and poster. Lastly, we agree with the reviewer that components of our curriculum intersect with various units of CIMER's Entering Mentoring curriculum but ultimately the goals of our curriculum differ. While CIMER focuses primarily on the mentor-mentee relationship, our course expands beyond that and extends to include interactions with course participants' peers, research group, department, and greater academic community.

Supplementary Materials: Lesson plans including materials, presentations, and links

REVIEWERS' COMMENTS:

Reviewer #1 (Remarks to the Author):

I appreciate the revision made to the manuscript. It is a stronger article now. Thank you for making these changes.

Reviewer #2 (Remarks to the Author):

The authors were very responsive to my critiques, and the manuscript is greatly improved. It should now be of broad interest to others hoping to develop similar courses.

It was important to include the evaluative comments from the chair, program director, and from postdoctoral fellows in the class. Not only did these show that the class was worthwhile, but they also brought out some of the possible hurdles that may need to be overcome for others to implement a similar course.

Also, the curriculum and resources in the supplemental pdf file will be very useful to anyone wanting to implement a similar course, will save them time in developing their course, and will allow them to learn from the authors.

We thank our reviewers and editor for their enthusiasm for our revised manuscript. The reviewers had no additional edits requested.